# Advancing Biodiesel Production System from Mixed Vegetable Oil Waste: A Life Cycle Assessment of Environmental and Economic Outcomes

**Farayi Musharavati** [1] , **Khadija Sajid** [2], **Izza Anwer** [3] , **Abdul-Sattar Nizami** [2,*], **Muhammad Hassan Javed** [2] , **Anees Ahmad** [2] and **Muhammad Naqvi** [4]

[1] Department of Mechanical and Industrial Engineering, Qatar University, Doha 2713, Qatar; farayi@qu.edu.qa

[2] Sustainable Development Study Centre, Government College University, Lahore 54000, Pakistan; khadijahsajid1998@gmail.com (K.S.); hassanjavedduggal@gmail.com (M.H.J.); aneesjamal180@gmail.com (A.A.)

[3] Department of Transportation Engineering and Management, University of Engineering and Technology, Lahore 54890, Pakistan; engr-izza@uet.edu.pk

[4] College of Engineering and Technology, American University of the Middle East, Egaila 54200, Kuwait; sayed.naqvi@aum.edu.kw

* Correspondence: asnizami@gcu.edu.pk

**Abstract:** This study aims to evaluate the environmental and economic performance of biodiesel production from mixed vegetable oil waste using the life cycle assessment (LCA) model. Due to its huge potential, Pakistan is taken as a case study. It produces 468,842 tons of vegetable oil waste annually. As no biodiesel production plant exists to process it, the environmental performance of biodiesel prototypes has not been investigated. Therefore, the current study is conducted to support the design of a plant to produce biodiesel from mixed oil waste. An attributional LCA was conducted using ReCiPe (H) and found that 400 kg of biodiesel can be produced from 1 t of mixed oil waste. The results, based on a functional unit of 1 ton, showed that biodiesel production from mixed vegetable oil waste is more eco-friendly than the existing landfilling practices with a global warming potential of $1.36 \times 10^{-4}$ kg $CO_2$ eq, human toxicity of 5.31 kg 1.4 DB eq, ozone depletion potential of 0.00271 kg CFC-11 eq, eutrophication potential of 0.0118 kg P eq, acidification potential of 123 kg $SO_2$ eq, and photochemical ozone formation of 51.4 kg $NO_x$ eq. Scenario modelling was conducted using electricity from photovoltaic solar cells, which decrease fine particulate matter formation from 44.5 to 0.725 kg $PM_{2.5}$ eq, instead of using electricity from a grid to the plant. Hotspot identification was carried out to highlight the effects of individual impact categories. An economic analysis showed that 638,839 USD/year revenue would be generated. Generating energy from discarded vegetable oils through biodiesel production presents a sustainable and economically viable approach. This process benefits the environment and contributes to cost savings by reducing waste disposal in landfills. Furthermore, it aligns with the principles of a circular economy, in which resources are reused and recycled. It also supports the pursuit of the United Nations' Sustainable Development Goals (SDGs), particularly SDG-7, which focuses on affordable and clean energy, and SDG-12, which emphasizes responsible consumption and production.

**Keywords:** vegetable oil; biodiesel production; waste-to-energy (WtE); life cycle assessment (LCA); sustainable development goals (SDGs)

## 1. Introduction

Pakistan generates 49.6 million tons of solid waste annually, with a yearly increase of 2.4% [1]. The government of Pakistan estimates that over 16,500 tons of municipal waste is generated every day, resulting in a weekly total of 87,000 tons of solid waste [2]. Additionally, the current solid waste management system is facing significant challenges due to

inadequate equipment, low public awareness, and a lack of urban planning. Thus, the lack of sound waste management practices creates serious environmental issues threatening the population's welfare and health. A total of 60–70% of the country's solid waste is collected, dumped, buried, or burned on vacant lots [3]. Reusing mixed vegetable oil waste instead of disposing it can create more mixed oil waste. This increase in the generation of such waste poses several challenges to its efficient management. Moreover, kitchens and food industries produce about 16.5 million tons of vegetable oil waste annually. This waste is usually disposed of in landfills and municipal solid waste or discharged into sewage systems [4].

At the start of the Industrial Revolution (the 20th century), energy utilization rapidly increased due to the increasing population and better living standards. In 2030, a 53% increase in global energy consumption is expected [5]. Currently, energy is primarily derived from natural gas (24%), coal (30%), and crude oil (33%), which are all fossil fuels [6,7]. The excessive use of non-renewable fossil fuels puts the energy security of people with limited access to these resources at risk, leading to climate change. Therefore, there is a need to find alternative energy sources to fossil fuels to guarantee energy security and tackle climate change [8].

In terms of economic development, energy is essential as it provides some necessary services to maintain the quality of human life and economic activity [9]. Pakistan's crude oil production in 2019 was 4.3 million metric tons, satisfying only 20% of the country's total petroleum needs. Alternatives to petroleum-based crude oil for diesel fuel are a major consideration. Hence, biodiesel production is gaining more attention as a direct replacement for crude oil petroleum as a blended component that is 100% renewable and biodegradable, as well as produces lower exhaust emissions compared to conventional diesel fuels [10]. In 2018–2019, Pakistan imported fossil fuels and imported 17.20 million tons of crude oil [3]. In this country, the electricity and transport sectors are the key users of fossil fuels. About 50% more energy is required for the transport and electricity sectors [11]. Moreover, Pakistan needs 10% blended biodiesel in fossil diesel by 2025 [12]. Therefore, lab-scale research on biodiesel production has frequently been conducted by organizations and universities in the country [1]. According to [13], various organizations in Pakistan have been developing biodiesel prototypes from diverse biomass sources. Moreover, many universities of the country have prepared biodiesel prototypes mainly from non-edible waste oils such as those from Jatropha seed oil.

Solid waste, including 4000 tons/day and 32.6 Mt/year of municipal solid waste (MSW), is appropriate for transformation into various waste-to-energy (WtE) forms [14]. Waste-to-energy is a process that reduces greenhouse gas emissions, recovers metals, and generates clean energy from waste materials. One form of material and energy recovery from waste is biodiesel production, mainly from mixed vegetable oil waste that reduces the burden on landfills and helps in energy recovery. Therefore, it is a secondary fuel that can manage various forms of urban and municipal waste, thereby improving waste handling [15]. In addition, waste is passed through a series of processes in which all the non-combustibles are removed for its production as follows: oil extraction, pretreatment, esterification, transesterification, and biodiesel refining, which can be applied after the process of purification to obtain biodiesel with a purity of 98% [16].

An optimized approach is used to check the sustainability of the process by performing a life cycle assessment (LCA), which is one of the most common sustainability assessment decision-making tools for assessing the impacts of different products or processes and environmental performance [17]. A product's life cycle starts with raw material extraction, then it is produced, transported, used, consumed, and disposed of, and finally its emissions and waste management are considered [18,19]. The LCA is a suitable tool for identifying a current project's environmental benefits and drawbacks and comparing them with those of conventional systems. Therefore, it helps policy and decision makers implement the process with minimal environmental impacts [20]. Environmental applications of LCA have increased worldwide over the last few years because it assesses the environmental impacts

of a current project throughout its life cycle. Although all processes result in resource consumption, emissions, and environmental impacts, an LCA looks at the process of the environment as a sink and a source and assesses the impact of different environmental impact categories such as human toxicity (HTP), global warming potential (GWP), ozone depletion potential (ODP), eutrophication potential (EP), and acidification potential (AP) [21]. In addition, it is the key factor for developing bioenergy support policies, including GHG savings, energy savings, and environmental and social acceptability [22].

This study examines the current generation and composition of mixed vegetable oil waste along with management practices and presents a design for the line production of biodiesel that can convert mixed oil waste into the formation of biodiesel. However, biodiesel production shows the potential and feasibility of biofuel as a substitute energy source and replacement for crude oil in the transportation sector. The goal is to reduce 10% of landfilled waste and 65% of MSW to be recycled by 2030. This study supports two UN Sustainable Development Goals (SDGs). Firstly, it aligns with SDG 7 for affordable and clean energy by showing how biodiesel from vegetable oil waste can be a practical and sustainable alternative to conventional energy. Secondly, our research supports SDG 12 for responsible consumption and production by recycling waste into energy, thus reducing environmental impacts and advancing sustainable practices. These two SDGs highlight the importance of our work, as emphasized in a recent editorial on sustainable energy research [23]. This study aims to evaluate a biodiesel production system's environmental and economic outcomes, utilizing mixed vegetable oil waste through a comprehensive LCA. This study thoroughly analyses material and energy flows and soil, water, and air emissions. The sustainability of the entire process was evaluated through an LCA using Gabi software (Version 10.0.0.71). A techno-economic analysis was also conducted to determine the economic feasibility of producing biodiesel for use as a fuel.

## 2. Materials and Methods

Section 2, 'Materials and Methods', is divided into four subsections: 'Section 2.1. Waste Characterization', 'Section 2.2. Study Design', 'Section 2.3. Life Cycle Assessment (LCA)', and 'Section 2.4. Economic Assessment'. These subsections provide information about the composition of mixed vegetable oil waste, the methodology of biodiesel production, the LCA's objectives and boundaries, and the financial aspects of biodiesel production, respectively.

The oil extraction phase includes different steps such as seed decortications, the filtering of oil, and the expulsion of oil from seeds as shown in Figure 1.

Moreover, the processes of transesterification and esterification, which need numerous inputs, such as catalysts in the form of the acid $H_2SO_4$, the alkali NaOH/KOH, alcohol in the form of methanol, and electricity.

### 2.1. Waste Characterization

In Pakistan, biodiesel from non-edible vegetable oil yields 0.09 thousand barrels daily. The physical characterization of the major components of mixed oil waste includes rapeseed oil (44.5%), castor oil (23.0%), waste cooking oil (11.5%), and Jatropha oil (21.0%). Moreover, out of every 100 tons of mixed oils processed, a portion is used to produce crude glycerol while the remainder can be used to make biodiesel.

### 2.2. Study Design

This study consists of a design for the line production of an extensive-scale biodiesel production plant from mixed vegetable oil waste as shown in Figure 2.

Soybean, palm, and peanut oils are vitally utilized in food industries and planted crops in Pakistan. These edible oils have enriched sources and a strong potential to produce biodiesel in large amounts. Thus, 1.25 of palm oil produces 1 L of biodiesel, and 1 L of biodiesel is produced from 1.3 L of soybean oil. Jatropha oil is the main source of biodiesel from Jatropha seeds in Africa and Asia. Jatropha seeds contain 30–35 wt.% oil which can

be converted into biodiesel. Waste cooking oil also has the highest capacity to produce biodiesel, and 89% of biodiesel is produced from WCO in the UK, similarly to many other countries.

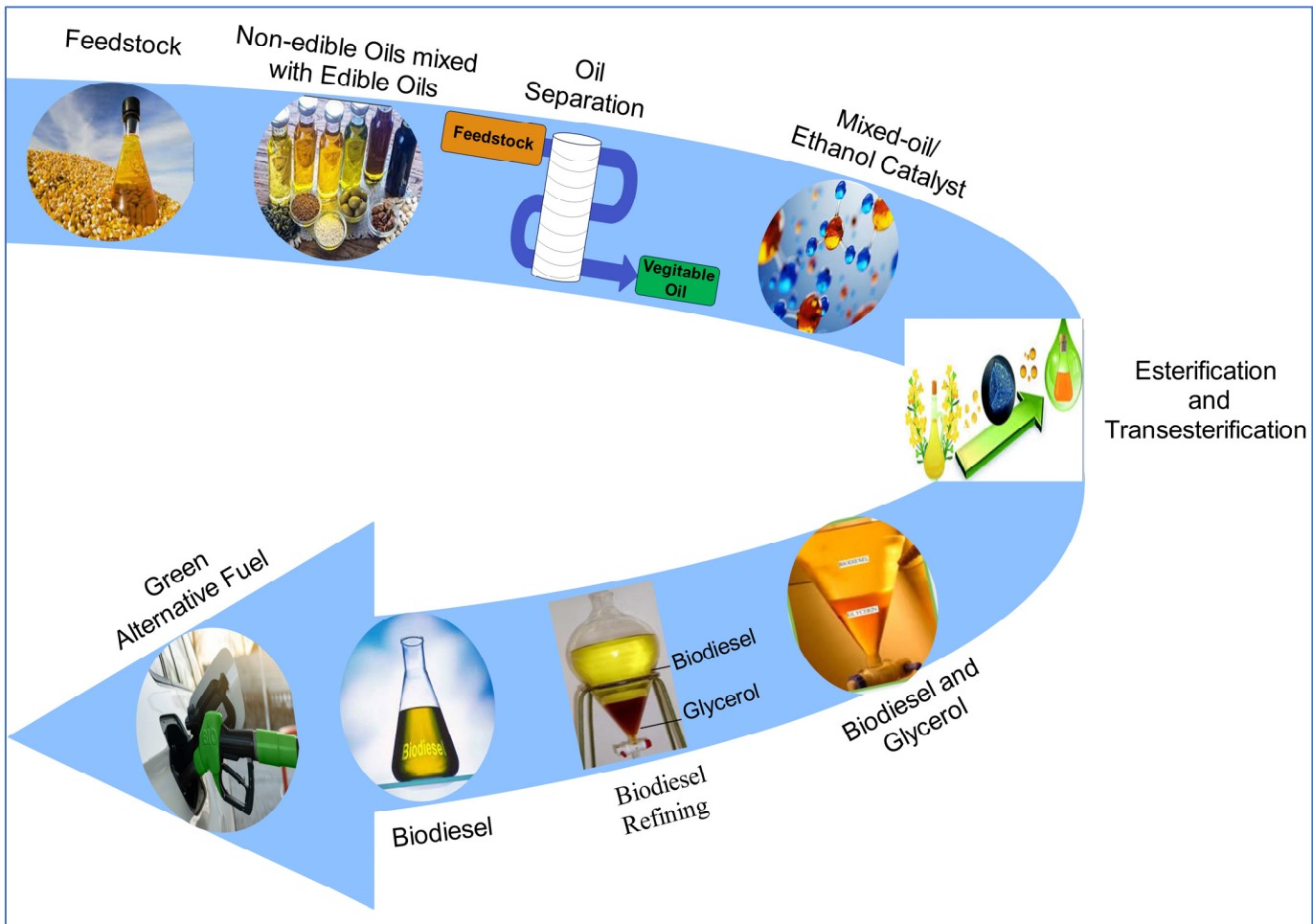

**Figure 1.** Schematic diagram of biodiesel production line.

Biodiesel production includes the following processes. The first step to produce biodiesel is the extraction of oil, in which the oil filtration, seed pressing, and decortication steps are carried out. The next step is the pretreatment, which is required to process feedstocks prior to their alteration into biodiesel. This step minimizes negative impacts on biodiesel production, such as suspended particles, polymers, FFAs, water, and gums. The pretreatment of oil has been shown to avoid soap formation during transesterification, eventually leading to an increased biodiesel yield. The third process involves two-step transesterification. Most non-edible oils have a higher content of FFAs from the pretreatment step; this amount must be reduced to 0.5–1% using an esterification reaction with an acid catalyst. The acid catalyst $H_2SO_4$ with methanol reduces the amount of FFAs. Hence, the transesterification process is carried out with the alkali catalysts NaOH or KOH with methanol. In this process, KOH is mostly preferred because of its low price, great productivity, and moderate yield. Therefore, KOH also decreases the oil's tendency to form soap. When KOH is used as a catalyst, it produces crude glycerol, and it is easier to separate this from the produced biodiesel using NaOH. Base transesterification produces a 98% biodiesel yield. Crude glycerol is generated as a byproduct that accounts for 10–12% of biodiesel. It is used as a processed industrial raw material that plays an important role in biodiesel chain sustainability and is the major bottleneck in producing biodiesel chains.

Therefore, crude biodiesel is subjected to wet washing to remove further impurities, such as catalysts, glycerol, soap, and residual alcohol, to obtain a purified biodiesel.

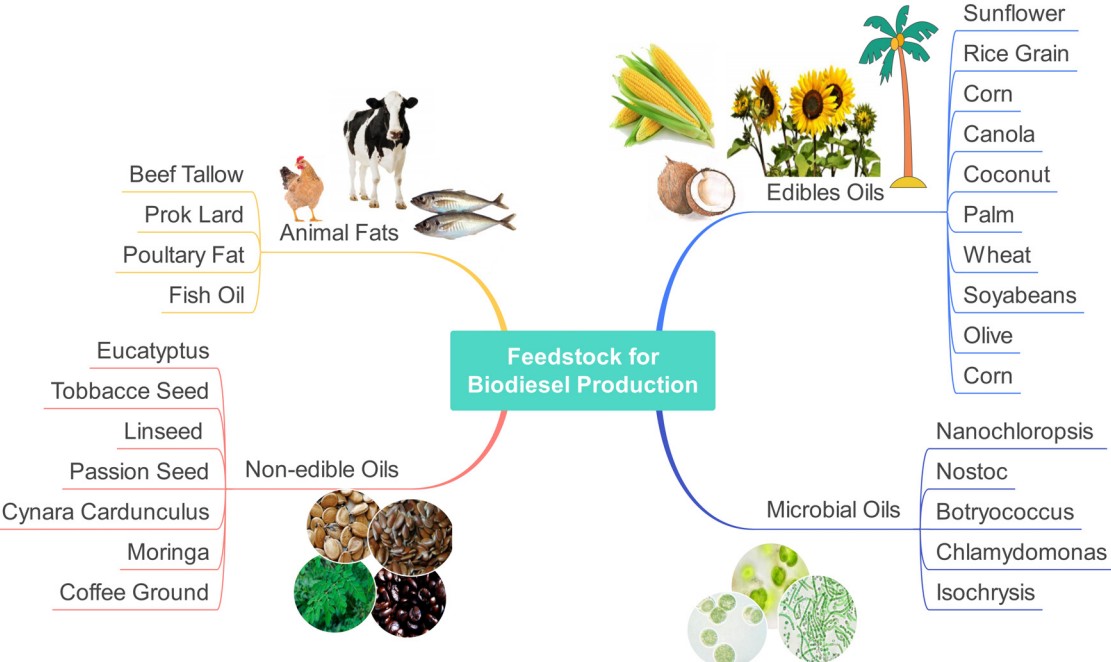

**Figure 2.** Various feedstock used to produce biodiesel.

### 2.3. Life Cycle Assessment (LCA)

An LCA is a method used to evaluate the sustainability of a product, process, or service by assessing its environmental impacts and benefits throughout its entire life cycle. Examining biodiesel production and consumption helps us to understand their ecological consequences. This study follows a standardized LCA approach in line with ISO 14040 guidelines and uses the Gabi LCA software (Version 10.0.0.71) as the primary tool for our analysis. The LCA process encompasses four fundamental stages: the goal and scope, life cycle inventory, life cycle impact assessment, and interpretation of the results, as outlined by [18,19].

#### 2.3.1. Goal and Scope

Our research aims to evaluate how producing biodiesel from various vegetable oils affects the environment, and this evaluation is conducted through an LCA. From an environmental standpoint, this study aims to determine the feasibility of producing biodiesel using various vegetable oils, following the methodological framework outlined in ISO as in [18,19]. One of the primary goals of the LCA process is to assess the environmental impacts associated with biodiesel production from diverse vegetable oil sources. However, this study evaluates the environmental assessment using an attributional LCA approach, which focuses on quantifying the hotspots and key environmental issues at various stages of biodiesel production. Moreover, system boundaries and functional units are both part of the goal and scope. Figure 3 shows the process of an LCA of biodiesel production.

#### System Boundaries

The scope of this study is gate-to-gate, along with system boundaries that consist of the following:

- The zero-burden assumption is selected, indicating that the biodiesel plant sources its inputs from various vegetable oils, including palm, soybean, castor, and waste cooking oils, which are all considered to have no environmental impact.

- The biodiesel production process considers both direct emissions generated on site and indirect emissions resulting from the use of electricity and fuel.
- The system boundary encompasses the collection of various vegetable oils from multiple sites and their transportation to the central facility. It extends through the biochemical treatment processes within a biodiesel production facility.

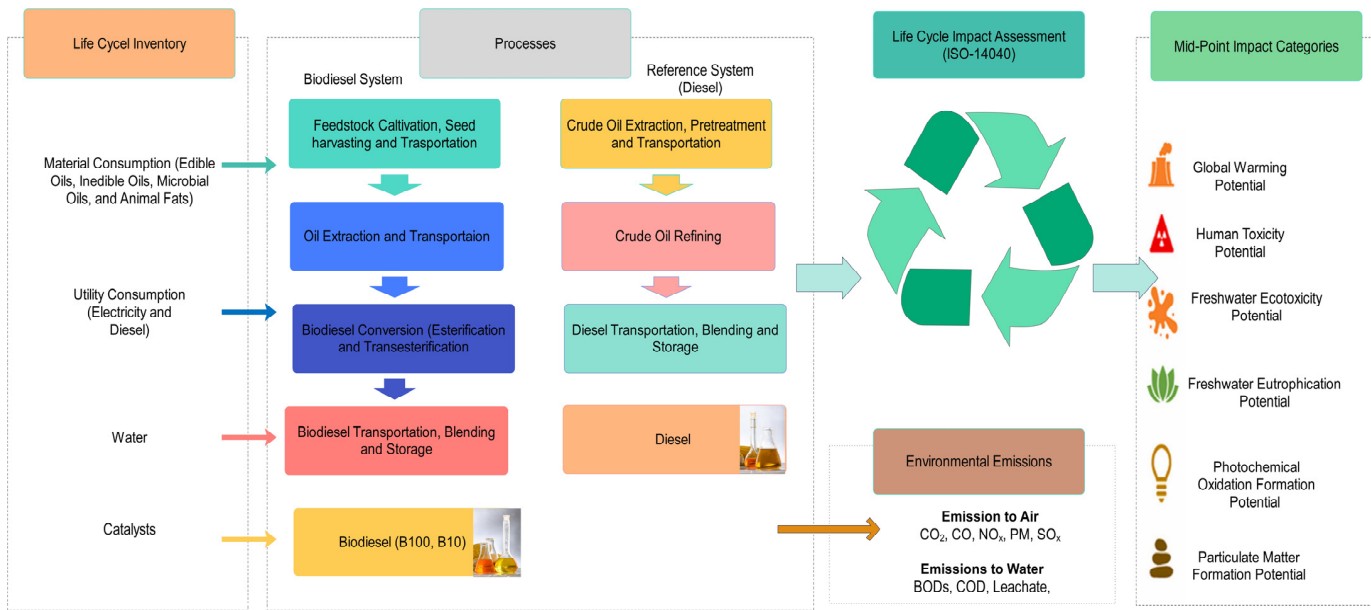

**Figure 3.** The life cycle assessment (LCA) of biodiesel production.

Functional Unit

The functional unit (FU) is a vital foundational standard for calculations and comparisons in LCA. One of the distinguishing features of LCA within environmental assessment methodologies is the selection of an FU. FU is an essential element in LCA analyses, allowing the comparison of results across different studies. It measures the system's function under study and establishes a reference point to normalize all inputs and outputs.

In the context of waste management systems, the choice of FU is closely linked to the system's inputs, goal and scope, and system boundaries, particularly how waste quantities are managed and processed. When assessing the environmental impact of biodiesel production from mixed vegetable oil waste sources, different FUs may be considered depending on specific factors like mass balance, transport distances, and energy considerations. In this study, the selected FU is 1 ton (1000 kg) of vegetable oil waste utilized in biodiesel production. All inputs and outputs in the analysis are standardized to this 1 ton functional unit, ensuring consistency and comparability across the study.

2.3.2. Life Cycle Inventory (LCI)

Life cycle inventory (LCI) is the second step in the analysis of LCA, which consists of measuring field data for which all the inputs and outputs of the system are considered and calculated. The data needed for biodiesel production include the production and composition of mixed vegetable oil, emission factors, fuel needs, and electricity requirements. In addition, the data obtained from the Gabi software serve as a basis for building the system model of the process. The system boundary, waste flow with mass balance, electricity consumption, and fuel requirements are shown in Figure 4.

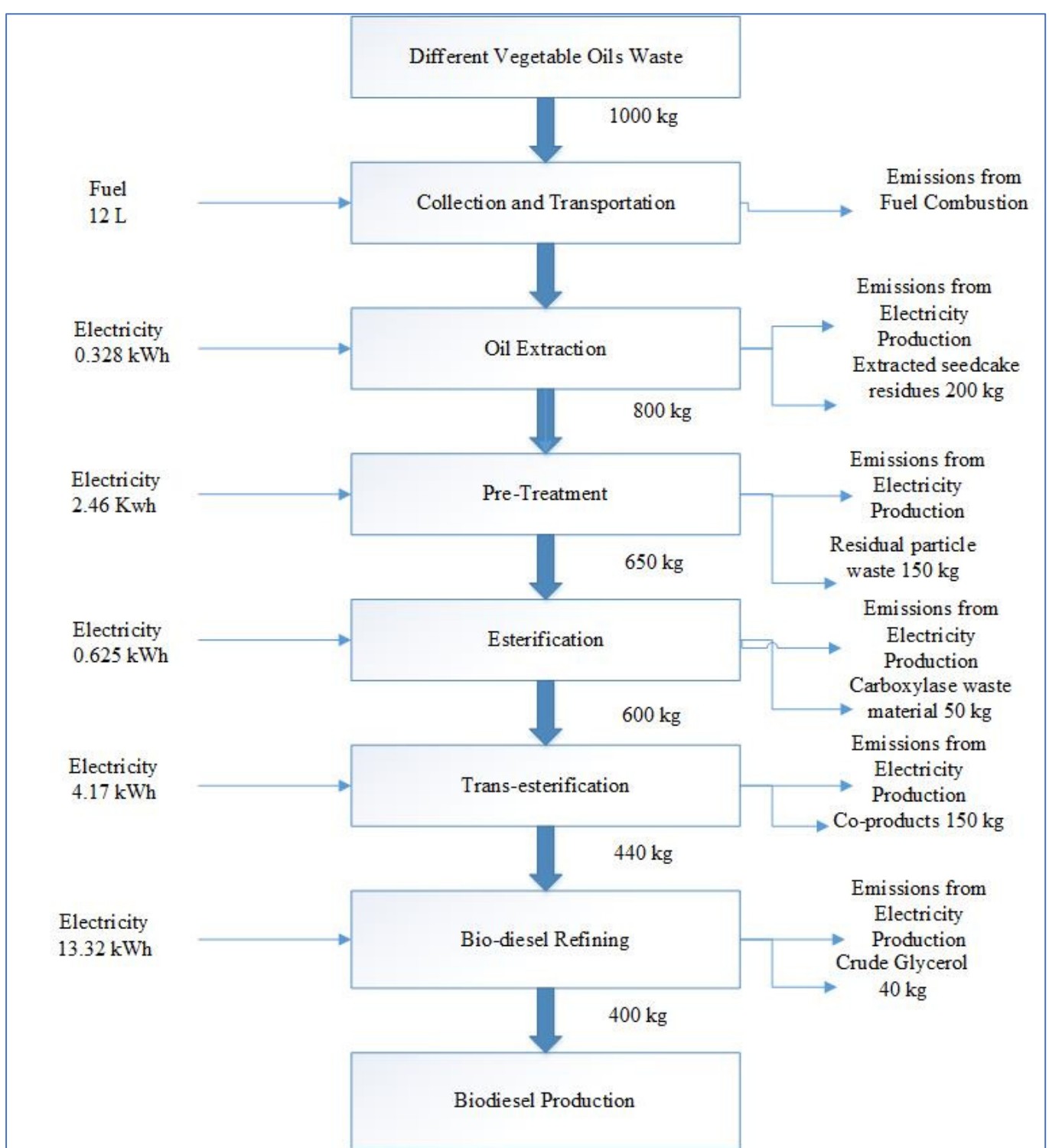

**Figure 4.** System boundary, vegetable oil waste stream flow with mass balance, electricity, and fuel consumption.

Table 1 presents a mass balance for the biodiesel production process. It lists the inputs, such as different oil wastes, amounting to 1 ton and energy consumption, which is measured in kWh per ton for electricity and litres for fuel. The final products include 400 kg of biodiesel, by-products such as metals and steam, and losses due to material losses and recycled waste. Additionally, the table attributes emissions into air, including $NO_x$ and $CO_2$, and emissions into water, such as $NO_3$, as a result of the production process.

**Table 1.** Mass balance for the biodiesel production process.

| Category | Quantities | Units | Amount |
|---|---|---|---|
| **Inputs** | | | |
| Material | Different Oils' Waste | ton | 1 |
| Energy Consumption | Electricity | kWh/ton | 20.278 |
| | Fuel | L | 12 |
| **Outputs** | | | |
| Main products | Biodiesel | kg | 400 |
| Other products | Metals | kg | 0.08 |
| | Steam | kg | 0.009 |
| | Material Losses | kg | 2.2 |
| Landfill | Recycled Waste | kg | 45.3 |
| Emissions into air | $NO_x$ | kg | 0.003405 |
| | $N_2O$ | kg | $2.94 \times 10^{-5}$ |
| | $CH_3$ | kg | $3.25 \times 10^{-7}$ |
| | $CO_2$ | kg | 0.00895 |
| | NMVOC | kg | 0.01 |
| Emissions into water | $NO_3$ | kg | $3.00 \times 10^{-5}$ |

### 2.3.3. Production and Composition of Mixed Vegetable Oil Waste

In Pakistan, 80% of the total consumption of edible vegetable oil stems from imports, while the remaining is produced locally. In 2022, the total consumption of edible vegetable oil, including palm, soybean, and peanut oils, was predicted to be approximately 3.9 million tons annually. However, domestic edible oil production was predicted to increase to 1 million tons in 2022. Palm oil accounts for a major portion of total domestic consumption (71%). Soybean oil accounts for 24% of total edible oil consumption.

LESCO (Lahore Electric Supply Company) is an electricity distributor in Lahore, Kasur, Okra, and Sheikhupura regions. The company's main objective is to ensure that the people in these areas can access a consistent and reliable electricity supply. To meet growing demand, LESCO uses a diverse mix of energy from various sources to supply electricity to the plant. According to [14], the breakdown of electricity sources is as follows: 29% hydro, 24% LNG, 14% furnace oil, 10% imported coal, 11% natural gas, 6% renewables, 4% nuclear, and 2% local. The electricity bills from WAPDA (Water and Power Development Authority) were assessed to determine the plant's power requirements for 2022. WAPDA has been assigned the duties of planning, executing, and investigating projects and schemes for generation, transmission, and distribution of power, as well as the water supply and prevention of water logging. This analysis calculated the electricity needs for each specific process within the plant. The plant operates for a total of 250 days a year, 5 days a week with 8 h of operation each day. On average, the plant consumes 20.27 kWh per ton of production.

### 2.3.4. Life Cycle Impact Assessment (LCIA)

Life cycle impact assessment (LCIA) is the most important stage of any LCA study. Many impact indicators, like inflows, outflows, mass, and energy consumption, are merged into LCIA to produce a single number for environmental performance. Human toxicity (carcinogenic and non-carcinogenic), photochemical ozone formation, resource depletion, acidification, ozone depletion, eutrophication, and global warming are some environmental impact categories. As a result, the four steps of the LCIA standard approach are normalization, weighting, characterization, and classification. However, ReCiPe (H) is the methodology utilized in the current study to evaluate the effects of LCIA.

### 2.4. Economic Assessment

The economic evaluation of this study takes operational costs and the returns on capital investment into account. Therefore, byproduct glycerol, material recovery, a reduction in

landfill costs, and scrap metals are all advantages of biodiesel production. Tangible and intangible benefits include preventing infections and odors from landfills and leachate poisoning of groundwater. Additionally, the annual cost is subtracted from the benefits received to determine the net benefit annually obtained from the method. Equation (1) is used to calculate the net benefits. In the meantime, Equations (2) and (3) compute the overall costs and benefits.

Economic indicators such as payback period (PB) and net present value (NPV) are used to determine a process's economic viability [24]. The difference between cash inflows and outflows of PV over a specific period of time is referred to as NPV. Moreover, it is a method used to determine a project's economic feasibility in terms of capital costs as well as the suitability of capital budgeting and investments.

Equations (1)–(3) are as follows.

$$\text{Net Benefits} = \text{Total Benefits} - \text{Total Cost} \tag{1}$$

$$\text{Total Benefits} = \text{Biodiesel} + \text{Metals} + \text{MR} \tag{2}$$

$$\text{Total Cost} = \text{LC} + \text{UC} + \text{EC} + \text{MC} \tag{3}$$

In this equation, MR stands for material recovery, LC for labor cost, UC for utility cost, EC for energy consumption cost, and MC for maintenance cost. However, as capital costs are one-time expenses for a project, they are not considered in Equation (3). Therefore, Equation (4) can be used to calculate NPV.

$$\text{NPV} = \sum\nolimits_{t=1} n \frac{C_t}{(1+k)t} - C_o \tag{4}$$

where, $C_t$ = Net cash inflow
$C_o$ = Capital cost
n = project lifespan
k = discount
t = time

The PP includes the net cash inflows generated from the initial investment because it is the expected time for the initial investment recovery. It is calculated via Equation (5).

$$\text{PP} = \frac{\text{Capital investment or initial investment cost}}{\text{Net Cash inflows(per year)}} \tag{5}$$

### 2.4.1. Life Cycle Costing (LCC)

Internal and external costs are considered in life cycle costing (LCC). Internal costs include the price of biodiesel l production, while external costs are the emissions during a process's life cycle stages [25]. Equation (6) is used to calculate LCC.

$$\text{Life cycle costing (LCC)} = \text{Internal cost (IC)} + \text{External cost (EC)} \tag{6}$$

### 2.4.2. Internal and External Costs

Operational and capital expenses are included in internal costs [26]. Maintenance, electricity, utility, and labor expenditures are all included in the operational costs of a plant. On the other hand, capital costs are associated with the plant's building, installation, equipment, and shipping. All associated costs of the current plants with similar facilities in other countries were also researched to obtain a reliable capital cost estimate. Equation (7) is used to determine the IC.

$$\text{IC} = C_l + C_u + C_r + C_m + C_M \tag{7}$$

where $C_l$, $C_u$, $C_r$, $C_m$, and $C_M$ stand for labor, utility, raw material, maintenance, and management costs, respectively.

External cost is the damage cost that is linked with environmental emissions. It can be calculated via Equation (8).

$$EC = \sum_{k=1}^{7} C_k \times E_{k,lc} \tag{8}$$

where $E_k$, $lc$ represents the emissions as determined via Gabi software. The coefficient values for $C_k$, which includes CO, $CO_2$, $NO_x$, $SO_2$, $CH_4$, PM, and NMVOC emissions, are obtained from the literature [21]. Table 2 shows the emissions values.

**Table 2.** Life cycle emissions and external cost coefficient of the biodiesel production process.

| Pollutants | Coefficient [a] | Emissions [b] |
|:---:|:---:|:---:|
| $CO_2$ | 44 | $5.85 \times 10^{-12}$ |
| $CH_4$ | 305 | 18.1 |
| CO | 828 | 10.3 |
| $SO_2$ | 7485 | 104 |
| $NO_x$ | 4712 | 49.7 |
| NMVOC | 2352 | 19.5 |
| PM | 8574 | 0.01209 |

[a] USD/ton; [b] Gabi (kg).

## 3. Results and Discussions

This section has four subsections, covering biodiesel production from mixed vegetable oil waste (Section 3.1), an environmental assessment (Section 3.2) of the impacts of biodiesel production through a life cycle approach, scenario modelling and assumptions (Section 3.3) considering different scenarios, and an in-depth economic evaluation (Section 3.4) of biodiesel production.

### 3.1. Biodiesel Yield and Properties

The present study's first step is identifying the percentage recovery of biodiesel based on the country's sources of mixed vegetable oil waste. Considering the quantity and composition of different types of vegetable oil production waste, a biodiesel production line for a large-scale plant is designed. The characterization and composition of different types of vegetable oil waste were taken from secondary data. The composition of mixed vegetable oil waste is mainly complex, and biodiesel is formed by a chain of hydrocarbons formed with two oxygen atoms, making it biologically active.

Biodiesel is an alternative fuel that can be derived from vegetable oils. Other alternative fuels include vegetable oil micro-emulsions, pyrolysis products of vegetable oils, and vegetable oils mixed with diesel oil. Methyl and ethyl esters can also be produced from vegetable oil or animal fat. In addition, biodiesel can also be produced from mixed vegetable oil, such as waste cooking oil, palm oil, Jatropha seed oil, castor oil, peanut oil, soybean oil, and algal oil [27]. Moreover, biodiesel can be used pure, as B100, or blended with diesel fuel with the blend denoted as BXX, in which XX is the biodiesel percentage in the blend. Hence, the most common ratio is B20, which is 80% diesel and 20% biodiesel [21].

One of the major benefits of biodiesel production is the low content of sulphur. In its chemical composition, oxygen is present, so its combustion is complete and reduces carbon monoxide, unburnt hydrocarbons, and particulate emissions, among with other contaminants [28]. Meanwhile, biodiesel can be used in any diesel engine without modifications and blended with fossil diesel in any proportion since they share similar properties [29]. Compared to fossil diesel, biodiesel has a lower calorific value of about 10% and performs worse at low temperatures. It also tends to solidify in extremely cold conditions, requiring

specific additives. In addition, a byproduct, glycerin, is obtained during its production process, which can be used in cosmetic and pharmaceutical industries after purification [30].

Biodiesel from different vegetable oil feedstocks is produced in many countries, including the USA, Brazil, Argentina, Thailand, Malaysia, Singapore, China, Indonesia, and India [28]. The USA produces 1.6 billion gallons of biodiesel, mainly from soybean (40%), canola (20%), palm (20%), and tallow (20%) oils. Brazil produces 6.8 million cubic meters of biodiesel from soybeans (80%), tallow (10%), and other vegetable oils (10%) [31]. China produces 2.43 billion liters of biodiesel from cooking vegetable waste (100%). India produces 185 million liters of biodiesel from cooking oil waste. The Philippines produces 203 million liters of biodiesel from coconut oil (100%) [6].

Some studies have shown that the calorific values of mixed vegetable oil seeds meet the energy demands to produce biodiesel. Non-edible vegetable oil results in a higher production of biodiesel. Jatropha seed oil has a calorific value of 37.27 KJ. Castor oil and cooking oil waste have calorific values of 35.50 KJ and 35.7 KJ, respectively [26]. On the other hand, edible vegetable oils, including palm, coconut, and jojoba oils, show calorific values of 37.30 KJ, 38.10 KJ, 39.86 KJ, respectively, and peanut and soybean oils have higher energy contents [32]. However, several methods exist in the literature for theoretically calculating biodiesel yields [33]. When calculating the theoretical amount of biodiesel that can be produced, it is assumed that one mole of mixed vegetable oil waste will yield three moles of biodiesel, with a 100% yield. However, the actual amount of biodiesel produced is determined using the density, volume, and molar weight. The density of mixed oils is calculated based on the weight and volume as follows:

$$\rho_{oil} = \frac{m_{oil}}{v_{oil}} \tag{9}$$

$\rho$ is the density, m is the mass, and v is the volume of oil.

Hence, the real amount produced by biodiesel is calculated by measuring the produced volume of oil, density of the biodiesel, and mass of biodiesel.

$$\frac{\rho_{biodiesel} \times V_{biodiesel}}{M_{biodiesel}} = \text{real amount of produced biodiesel} \tag{10}$$

Therefore, to obtain a 100% yield of biodiesel, the equation is as follows:

$$n = \frac{\text{real amount of produced biodiesel}}{\text{therotical amount of biodiesel}} \tag{11}$$

The benefits of the production of biodiesel include a decrease in the environmental impacts of MSW, its stable thermal and energy properties, a reduction in the share of landfilled waste, lower leachate production, the conservation of fossil fuels, decreased GHG emissions, and lower pollutant content [34]. However, despite its benefits, its challenges include high capital costs, landfilling options, unstable market conditions, and the availability of industries for co-combustion [1]. Meanwhile, another application of biodiesel production is in the transportation sector because it is a clean, renewable, and biodegradable alternative to conventional fossil diesel. Biodiesel produces fewer pollutants and a lower carbon output than other petroleum products. Compared to petroleum diesel, biodiesel produces less sulfur dioxide, fewer soot particles, and fewer unburnt hydrocarbons. By using biodiesel, people can realize health benefits because they need to spend less on healthcare products. Moreover, biodiesel can also be used to generate energy and electricity and provide heat. Nowadays, the concept of bioheat has continued to grow and depend on biofuels such as biodiesel being used as a source of energy [21]. In addition, a comparison of different studies for the line production of biodiesel is given in Table 3. Through different assessment models, the life cycle of biodiesel production is assessed by using different system inputs and outputs with different system boundaries and functional units. It is as follows.

**Table 3.** Different studies on the environmental impact assessment of biodiesel production.

| System Boundaries | System Outputs | Functional Unit | Assessment Method | Highlights | Reference |
|---|---|---|---|---|---|
| Gate-to-gate, waste pretreatment, oil extraction, esterification, transesterification, and commercial biodiesel transportation | Waste (solid and liquid), heat, electricity biodiesel, and glycerol | 1 ton | ReCiPe 2016 | Biodiesel production is promising, leading to lower levels of carbon dioxide | Present study |
| Cradle-to-gate, oil waste collection, and oil waste pretreatment | Potassium sulfate, distillation residue, and wastewater | 1 ton | ReCiPe 2008 | Emissions from combustion and certain chemicals are major environmental issues in biodiesel production. | [35] |
| Gate-to-gate, pretreatment, transesterification, biodiesel washing, biodiesel dehydration, crude glycerol neutralization, and methanol recovery | Biodiesel, glycerol, electricity, heat, phosphate, free fatty acids, and dipotassium | 1 MJ | IMPACT 2002+ | Sensitivity analysis showed neutralizing crude oil, consumption of electricity, and methanol impact human health and climate change | [36] |
| Cradle-to-gate, waste collection, and transportation | Biodiesel | 1 ton | CML | Transesterification by alkali catalyst contributes to environmental burdens. | [37,38] |
| Cradle-to-grave, fresh oil production, pretreatment, collection, transportation, combustion, and transesterification | Biodiesel, methanol, and glycerol | 1 kg | - | Due to sulfur compounds in WCO and contaminants released during the cooking process, the resulting biodiesel produces more sulfur dioxide emissions than diesel. | [39] |
| Cradle-to-cradle, pretreatment, collection, transportation, oil, and transesterification process | Gas emissions (waste), heat, wastewater, glycerol, and biodiesel | 1 ton | CML with all categories and Eco-indicator 99 | Transesterification process had a significant environmental impact due to increased electricity consumption. | [40] |
| Oil waste collection and transportation, waste esterification, transesterification, and pure biodiesel | Waste disposal, glycerol, and biodiesel | 1 kg | Eco-indicator 99 | When a territory is small, centralized production is more eco-friendly, but as the territory increases, decentralization becomes more advisable. | [41,42] |

*3.2. Environmental Assessment Using LCA*

In this context, the study's system boundary is gate-to-gate, and the FU is set at 1 ton of mixed vegetable oil waste. Our analysis of the results employs the ReCiPe Midpoint (H) LCIA method. The data on the quantity, materials, fuel, and energy consumption of the mixed vegetable oil waste were collected from the published literature on the

country's overall production of biodiesel [43]. However, the other data (the electricity emissions) were taken from the database. Hence, the LCIA results are converted into impact categories (classification). The classified results are collected for each category indicator (characterization). The ReCiPe methodology used in this study is the follow-up of CML 2002 and Eco-indicator 99 methodologies. The indicator scores are measured in the same way as the Eco-indicator 99 methodology, and this approach uses both midpoint and endpoint modelling [11]. In addition, it is a coordinated LCIA method at a midpoint level [23]. It covers 18 midpoint impact categories, including ozone formation (terrestrial ecosystems, human health), ozone depletion, marine ecotoxicity, freshwater ecotoxicity, human toxicity (cancer, non-cancer), terrestrial acidification, terrestrial ecotoxicity, land use and water consumption, fossil depletion, climate change, ionizing radiation, resource depletion, fine particulate matter formation, and marine eutrophication. The endpoint indicators are mainly grouped into three main categories: ecosystems, resources, and human health.

### 3.2.1. Midpoint Assessment

The environmental categories in which the substances are released into the environment are determined by the changes made to the natural environment and are included in the midpoint assessment. These are also known as impact categories. Table 4 summarizes the midpoint results for biodiesel production. Overall, our findings showed that biodiesel production has huge potential to enhance the environment in terms of all effect categories. The highest reduction is found in global warming potential ($1.36 \times 10^{-4}$ kg $CO_2$ eq.), fossil depletion ($3.29 \times 10^{-3}$ kg oil eq.), ozone depletion (0.00271 kg CFC-11 eq.), and all ecotoxicity impacts (freshwater: 0.647 kg 1.4 DB eq., freshwater eutrophication: 0.0118 kg P eq., marine eutrophication: 0.134 kg N eq., and marine ecotoxicity: 9.07 kg 1.4 DB eq.). The following categories are covered under the LCIA: photochemical ozone formation, ozone depletion, human toxicity, ecotoxicity (terrestrial, freshwater, and marine), the depletion of fossils, acidification, the potential for global warming, eutrophication (marine and freshwater), ionizing radiation, resource depletion, and particle formation. The findings of the biodiesel production's midpoint assessment are shown in Table 4.

**Table 4.** Midpoint assessment of biodiesel production.

| Impact Categories | Unit | Values |
|---|---|---|
| Climate change, default, excl. biogenic carbon | kg $CO_2$ eq. | $1.36 \times 10^4$ |
| Fine particulate matter formation | kg PM2.5 eq. | 44.5 |
| Fossil depletion | kg oil eq. | $3.29 \times 10^3$ |
| Freshwater consumption | $m^3$ | 326 |
| Freshwater ecotoxicity | kg 1.4 DB eq. | 0.647 |
| Freshwater eutrophication | kg P eq. | 0.0118 |
| Human toxicity, cancer | kg 1.4-DB eq. | 5.31 |
| Human toxicity, non-cancer | kg 1.4-DB eq. | $1.29 \times 10^3$ |
| Ionizing radiation | kBq Co-60 eq. to air | 27.4 |
| Land use | Annual crop eq. per year | 287 |
| Marine ecotoxicity | kg 1.4-DB eq. | 9.07 |
| Marine eutrophication | kg N eq. | 0.134 |
| Metal depletion | kg Cu eq. | 2.8 |
| Photochemical ozone formation, ecosystems | kg $NO_x$ eq. | 51.5 |
| Photochemical ozone formation, human health | kg $NO_x$ eq. | 51.4 |
| Stratospheric ozone depletion | kg CFC-11 eq. | 0.00271 |
| Terrestrial acidification | kg $SO_2$ eq. | 123 |
| Terrestrial ecotoxicity | kg 1.4-DB eq. | $1.2 \times 10^4$ |

Biodiesel production contributes to a decrease in pollutant emissions without causing an increase in greenhouse gas (GHG) emissions. Consequently, several studies have indicated that burning can elevate the concentration of $CO_2$ in the atmosphere. However,

carbon absorption throughout a plant's life cycle can offset this increase in emissions [33]. The GWP of the biodiesel process is $136 \times 10^{-4}$ kg $CO_2$ eq. Eutrophication (freshwater or marine) is aquatic nutrient enrichment brought on by $H_3PO_4$ and $PO_4^{3-}$; it causes environmental deterioration. The NP is measured in kg P or N eq. and primarily from landfill or diesel emissions. As a result, open burning emits pollutants into the atmosphere, which eventually settle with rain [44]. Emissions from sedimentation raise the productivity and nutrient levels in water bodies. Thus, algae absorb nutrients that are needed by other aquatic organisms [45]. Eventually, they are decomposed by bacteria and all die. This situation leads to a decrease in the level of DO as the amount of oxygen available to living aquatic organisms decreases [8]. The marine eutrophication of the process is 0.134 kg N eq., and the freshwater eutrophication is 0.0118 kg P eq.

The production of biodiesel is a feasible option for reducing the potential for terrestrial acidification because it results in fewer emissions of $NO_x$ and $NH_3$ during the processing stage. It is expressed in kg $SO_2$ eq. as the unit of terrestrial acidification. The decrease in emissions harms plant and animal life and causes the acidity of soil or aquatic ecosystems to decrease. Thus, one effective method for reducing the potential effects of terrestrial acidification is the manufacture of biodiesel. In addition, the terrestrial acidification potential of biodiesel was 123 kg $SO_2$ eq. Human toxicity is classified into effects that cause cancer and effects that do not cause cancer, and it is related to the maximum daily intake for human toxicity. It is mainly caused by heavy metals, hydrogen sulfide, nitrogen oxides, and formaldehyde and is measured in kg 1.4 DB eq [46]. In the current study, the human toxicity potential (non-cancer) was $1.29 \times 10^{-3}$ kg 1.4 DB eq and the HTP (cancer) was 5.31 kg 1.4 DB eq.

Ozone layer depletion causes damage to human health and ecosystems. However, more Ultraviolet B (UVB) radiation is now at the Earth's surface, which is bad for ecosystems and human health. Natural elements, including methane, water, nitrogen dioxide, and halogenated components, are the main contributors to ozone depletion [47]. Ozone depletion has significantly impacted the ecosystem. Thus, the industrial use of very stable halocarbon gases has led to the formation of halogen compounds in the stratosphere. These gases are found in landfills, and their presence poses a sepulcher environmental threat [48]. The ozone depletion potential of biodiesel was calculated as 0.00271 kg CFC-11 eq.

Compounds that are reactive in the atmosphere and the photochemical ozone formation process can harm human health and the environment. Moreover, various volatile organic compounds (VOCs) produced by activities; the use solvents and motor vehicles are significant sources of this type of pollution. The main contributors to its creation are $NO_x$, NMVOC, and $CH_4$, while $NO_x$ is generated during transportation. The value of photochemical ozone formation for human health is 51.4 kg $NO_x$ eq. The particles released into the atmosphere are referred to as the particulate matter formation. PM10 is the term for any organic and inorganic compounds with a diameter of less than 10 m, such as $SO_x$, $NO_x$, $NH_3$, and VOCs [49]. It negatively affects health, leading to respiratory disorders. Therefore, because it necessitates more energy-intensive waste collection and treatment processes, the landfill is the least preferable alternative in terms of air pollution [50]. The PMF in this study was 44.5 kg $PM_{2.5}$ eq. Ecotoxicity refers to the effect of toxic substances on wetland ecosystems and forests. The effect on oceans is known as marine ecotoxicity, while the effect of toxic substances on freshwater bodies like rivers and lakes is known as freshwater ecotoxicity [51]. According to the current study, the marine and freshwater ecotoxicity values were about 9.07 and 0.647 kg 1.4 DB eq, respectively.

Ionization is the environmental release of radioactive elements that cause a higher radiation potential. A significant amount can result in immediate fatalities, severe radiation burns, or acute consequences. This is a result of the radioactive elements found in rocks and soils of landfills [7]. The ionizing radiation potential of the current model was 27.4 kBq Co-60 eq. to air. Resource depletion is the consumption of natural resources. The electricity consumption for 1 t biodiesel production is 21.75 kWh. Studies indicate a decrease in fossil fuels that are mostly used in the power sector [52]. The generation of biodiesel was

$3.29 \times 10^{-3}$ oil eq., which is essentially no fossil depletion. The electricity generated from biodiesel can balance out the electricity used to produce biodiesel [32]. The metal depletion potential was 2.8 kg Cu eq.

While LCAs have been used in South Africa, India, Russia, and Brazil over the past 15 years, they have also been extensively used in several European nations [53]. The production line for mixed vegetable oil waste biodiesel, its composition, and its percentage recovery affects the fuel's economic and environmental advantages. Biodiesel has many uses, and its manufacturing characteristics vary depending on location [17]. Furthermore, the direct emissions of WtE facilities and the LC performance of biodiesel are the waste composition, recovery efficiency, and type of biological treatment. Since biodiesel may be used as a substitute for petroleum diesel, it boosts energy security, improves the environment and air quality, and uses less energy during production than conventional fossil fuels [54]. Therefore, decreased eutrophication and acidification are brought on by the reduction in $NO_x$ emissions [55]. The current study carried out an LCA of the generation of biodiesel from mixed vegetable oil waste.

### 3.2.2. Normalized Results

The environmental category units are different for each category. Hence, they cannot be compared. The results are normalized, in which the category indicators are divided by a reference value. Moreover, normalized results signify the average environmental impact that a single statistical person exerts, and they are expressed in person equivalent (PE) units [17]. The normalized results are shown in Table 5. The current research paper utilizes the reCiPe 2016 V1.1 (H), global (PE) eliminating biogenic carbon, midpoint normalization built-in Gabi program. Human toxicity (non-cancer), ozone formation, and particulate matter formation have proportionately bigger contributions to the production of biodiesel, while ecotoxicity and climate change have moderate effects. The impacts of land use on eutrophication are minimal.

**Table 5.** Normalized LCIA results of the biodiesel production process.

| Categories | Unit | Values |
|---|:---:|:---:|
| Ecosystems | | |
| Climate change freshwater ecosystems | species. yr | $4.17 \times 10^{-7}$ |
| Climate change terrestrial ecosystems | species. yr | 0.0153 |
| Freshwater consumption, freshwater ecosystems | species. yr | $2.27 \times 10^{-7}$ |
| Freshwater consumption, terrestrial ecosystems | specie. yr | 0.0016 |
| Freshwater ecotoxicity | species. yr | $1.8 \times 10^{-7}$ |
| Freshwater eutrophication | species. yr | $3.17 \times 10^{-6}$ |
| Land use | species. yr | 0.00102 |
| Marine ecotoxicity | species. yr | $3.81 \times 10^{-7}$ |
| Marine eutrophication | species. yr | $8.84 \times 10^{-8}$ |
| Photochemical ozone formation, ecosystems | species. yr | 0.00266 |
| Terrestrial acidification | species. yr | 0.0104 |
| Terrestrial ecotoxicity | species. yr | $5.49 \times 10^{-5}$ |
| Human Health | | |
| Climate change human health | DALY | 3.8 |
| Fine particulate matter formation | DALY | 8.39 |
| Freshwater consumption, human health | DALY | 0.184 |
| Human toxicity, cancer | DALY | 0.00529 |

**Table 5.** *Cont.*

| Categories | Unit | Values |
|---|---|---|
| Human Health | | |
| Human toxicity, non-cancer | DALY | 0.0886 |
| Ionizing radiation | DALY | $6.98 \times 10^{-5}$ |
| Photochemical ozone formation, human health | DALY | 0.014 |
| Stratospheric ozone depletion | DALY | 0.000431 |
| Resources | | |
| Metal depletion | $ | 420 |
| Fossil depletion | $ | $7.4 \times 10^{-4}$ |

### 3.2.3. Hotspot Identification

To regulate the main contributors' stages in the life cycle of biodiesel production, a comparison of the contributions of the individual processes of biodiesel production is shown in Figure 5. This shows a comparison between the involvement of individual processes and landfills. The impact of oil extraction on the overall impact categories is minimal and hence considered negligible. The impact of the pretreatment stage on all the impact categories is very low and almost the same for each. However, the impacts of the esterification and transesterification processes on freshwater, terrestrial, and marine ecotoxicity are almost the same and account for <20%. The stage of biodiesel refinery has the highest contribution, primarily on the global warming potential, human toxicity potential, eutrophication, and acidification potential. In addition, along with the refining process, landfills account for 45% of human toxicity and marine ecotoxicity. The major reason for their higher impacts is the consumption of electricity and heat. Therefore, these two processes are more fuel-intensive than those in the biodiesel production system. The modelling results highly stimulate the data on electricity consumption. Electricity is a cleaner technology, but the emissions from it are at the time of production. The electricity supplied to a plant by LESCO is from the grid mix, and it mostly involves non-renewables. Our overall results show that the impacts are the same across the categories considered. Hence, the system boundary of the present study is from gate to gate; only biodiesel-obtaining processes were considered. Thus, for a plant, only collection and transportation are considered. Another study also shows the usage of electricity and transportation as the dominant stages [4].

### 3.2.4. Endpoint Assessment

The term "endpoint assessment" (also known as "damage categories") refers to how much of a material is released into the environment before it causes harm. These categories also cover the environment. Endpoint indicators combined all effect subcategories into three major categories: ecosystems, human health, and resources. The results of the biodiesel production of our endpoint assessment are shown in Table 6.

### 3.3. Scenario Modeling and Assumptions

The midpoint results of the LCA of the current model (Figure 6) show that the electricity supplied to the plant by LESCO is from a grid mix, in which the major contribution is from non-renewables. Therefore, all the secondary emissions of electricity are considered in the LCA. However, in scenario modelling, the electricity supply is assumed to be from photovoltaic solar cells instead of the grid mix. A comparison of scenario modelling and the current model is shown in Figure 6.

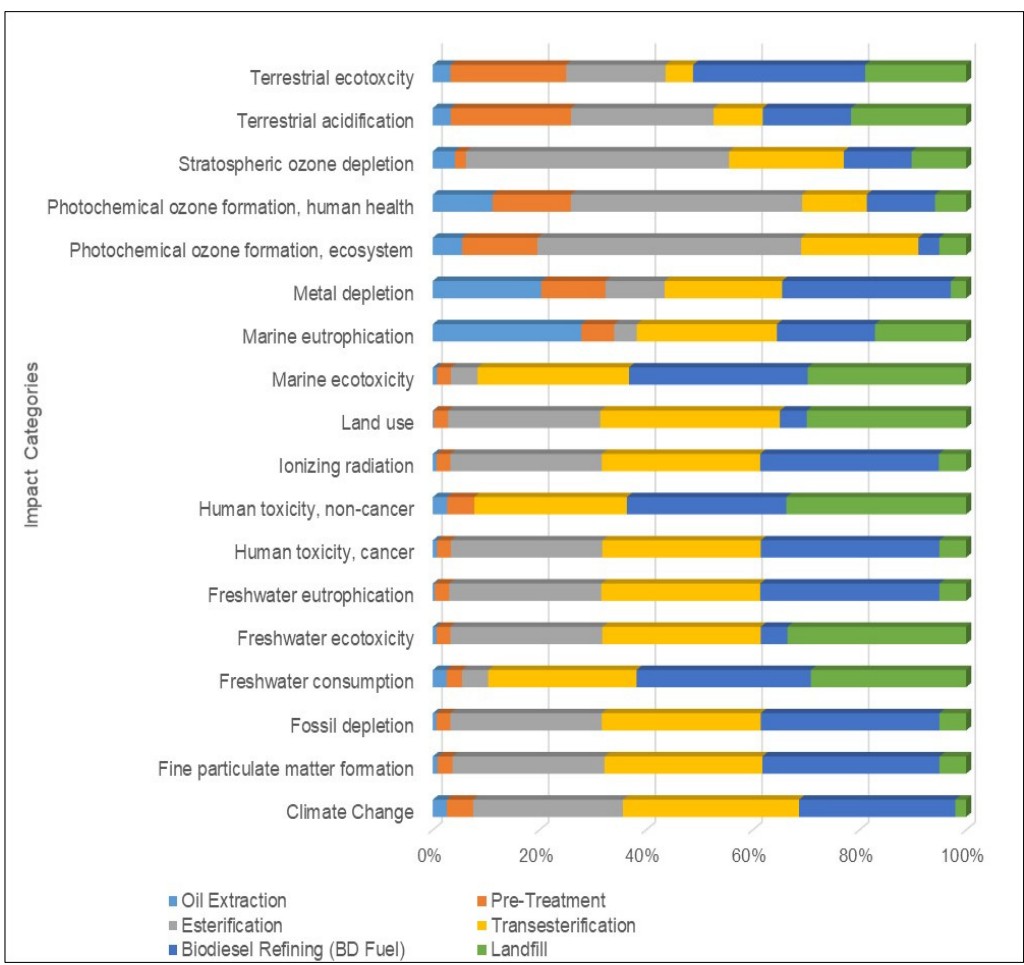

**Figure 5.** Role of individual processes in the overall environmental impact of the biodiesel production process to midpoint categories linked to landfill emissions.

**Table 6.** Endpoint results of life cycle characterization of the biodiesel production line.

| Categories | Unit | Values |
|---|---|---|
| Ecosystems | | |
| Climate change terrestrial ecosystems | species. yr | $3.82 \times 10^{-5}$ |
| Climate change freshwater ecosystems | species. yr | $1.04 \times 10^{-9}$ |
| Photochemical ozone formation, ecosystems | species. yr | $6.64 \times 10^{-6}$ |
| Freshwater consumption, freshwater ecosystems | species. yr | $5.68 \times 10^{-10}$ |
| Freshwater consumption, terrestrial ecosystems | species. yr | $4.01 \times 10^{-6}$ |
| Land use | species. yr | $2.55 \times 10^{-6}$ |
| Marine ecotoxicity | species. yr | $9.52 \times 10^{-10}$ |
| Marine eutrophication | species. yr | $2.21 \times 10^{-10}$ |
| Freshwater ecotoxicity | species. yr | $4.5 \times 10^{-10}$ |
| Freshwater eutrophication | species. yr | $7.93 \times 10^{-9}$ |
| Terrestrial acidification | species. yr | $2.61 \times 10^{-5}$ |
| Terrestrial ecotoxicity | species. yr | $1.37 \times 10^{-7}$ |

**Table 6.** *Cont.*

| Categories | Unit | Values |
|---|---|---|
| *Human health* | | |
| Climate change, human health | DALY | 0.0127 |
| Human toxicity, cancer | DALY | $1.76 \times 10^{-5}$ |
| Human toxicity, non-cancer | DALY | 0.000295 |
| Fine particulate matter formation | DALY | 0.028 |
| Freshwater consumption, human health | DALY | 0.000614 |
| Ionizing radiation | DALY | $2.33 \times 10^{-7}$ |
| Stratospheric ozone depletion | DALY | $1.44 \times 10^{-6}$ |
| Photochemical ozone formation, human health | DALY | $4.67 \times 10^{-5}$ |
| *Resources* | | |
| Fossil depletion | $ | 247 |
| Metal depletion | $ | 1.4 |

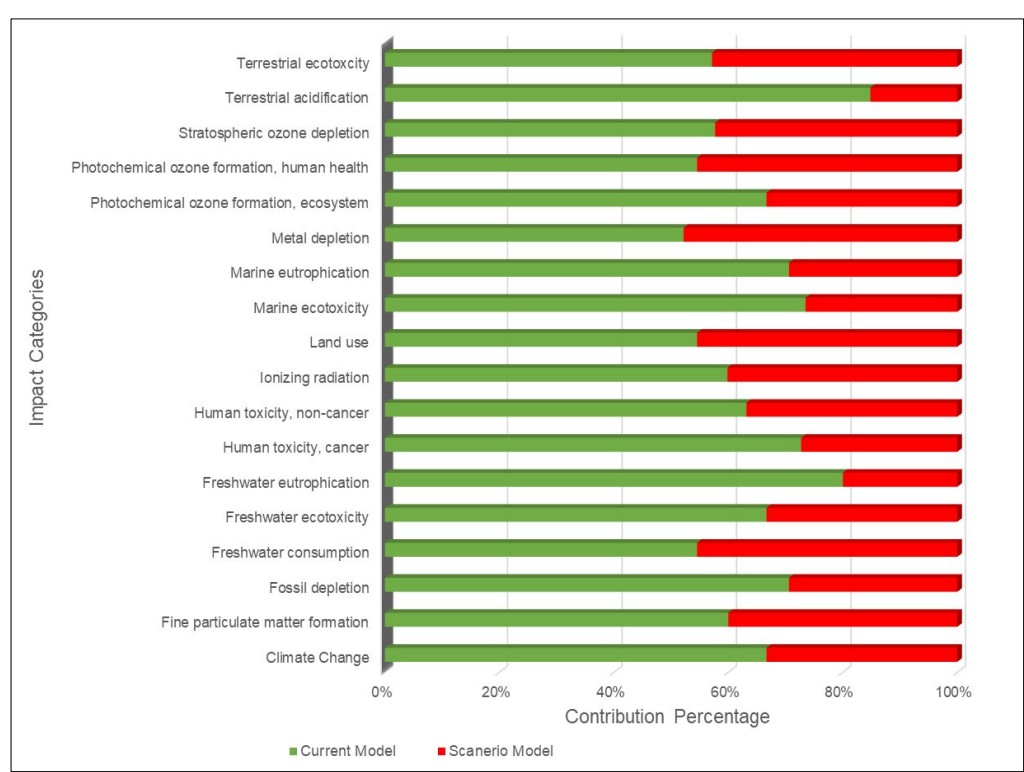

**Figure 6.** Comparison of current model versus scenario model of biodiesel production process. (In the current model, electricity is supplied from the grid mix, while in the scenario model, it is supplied from photovoltaic solar cells).

Thus, it is apparent that the fine particle matter formation decreases from 44.5 to 0.725 kg PM$_{2.5}$ eq., and the fossil depletion increases from $3.29 \times 10^{-3}$ to 196 kg oil eq. The effect on freshwater consumption in the current model is 326 m$^3$, while it slightly decreased to 279 m$^3$ in the scenario model. Similarly, the terrestrial acidification decreases from 123 to 2.15 kg SO$_2$ eq., and human toxicity and cancer decrease from 5.31 to 0.405 kg 1.4 DB eq., respectively. In addition, the effect on metal depletion increases from 2.8 to 12.6 kg Cu eq. The midpoint results for the biodiesel production of the scenario model are in Table 7.

**Table 7.** Midpoint results of life cycle characterization of biodiesel production line of the scenario model.

| Impact Categories | Unit | Values |
|---|---|---|
| Climate change, default, excl. biogenic carbon | kg $CO_2$ eq. | 654 |
| Fine particulate matter formation | kg PM2.5 eq. | 0.725 |
| Fossil depletion | kg oil eq. | 196 |
| Freshwater consumption | $m^3$ | 279 |
| Freshwater ecotoxicity | kg 1.4 DB eq. | 0.136 |
| Freshwater eutrophication | kg P eq. | 0.00111 |
| Human toxicity, cancer | kg 1.4-DB eq. | 0.405 |
| Human toxicity, non-cancer | kg 1.4-DB eq. | 194 |
| Ionizing radiation | kBq Co-60 eq. to air | 8.05 |
| Land use | Annual crop eq. per year | 30.2 |
| Marine ecotoxicity | kg 1.4-DB eq. | 4.33 |
| Marine eutrophication | kg N eq. | 0.00765 |
| Metal depletion | kg Cu eq. | 12.6 |
| Photochemical ozone formation, ecosystems | kg $NO_x$ eq. | 2.68 |
| Photochemical ozone formation, human health | kg $NO_x$ eq. | 2.62 |
| Stratospheric ozone depletion | kg CFC-11 eq. | 0.000159 |
| Terrestrial acidification | kg $SO_2$ eq. | 2.15 |
| Terrestrial ecotoxicity | kg 1.4-DB eq. | $1.33 \times 10^4$ |

Biodiesel production from mixed vegetable oil waste reduces the problems of the waste disposal/handling of waste, reduces emissions by avoiding them, and provides economic benefits. There are some weaknesses in current SWM practices, such as waste handling and the local fleet industry for handling waste. The lack of coordination among stakeholders, including municipalities and the informal sector, treatment technology and management, and initial capital investments are the main difficulties in implementing a sound SWM model [56]. Moreover, there is a dire need to improve the SW sector through proper stakeholder management and coordination. In addition, economic incentives should be given to level up applicable enterprises and implement a sound solid waste management model [57].

Biodiesel produces a clean-burning, renewable alternative fuel to conventional petroleum-based fuels [58]. It improves energy balance and security. Thus, locally manufactured biodiesel can be directly substituted for conventional diesel fuels. Biodiesel produced from soybeans has a positive energy balance that shows a higher yield in a higher amount of energy for every unit of fossil fuel consumed [59]. Moreover, biodiesel production also reduces emissions and improves air quality because of its lower life cycle rating and overall lower emissions which improve air quality. Other biodiesel applications include its use as fuel filters, in oil spill cleanups, as heating oil, and in biodiesel electricity generators.

In Pakistan, vegetable oil is mainly used to treat biodiesel, either with ethanol or methanol to synthesize it. The basic reason for using methanol worldwide is its lower price; coal is the main production source. Around 180 billion tons of coal reserves are in Pakistan, the fifth largest in the world. However, in Pakistan, ethanol production is also higher because its 76 operational sugar mills produce 300,000 tons of cane daily. Some distillery units have a capacity to produce 2 million tons of molasses to form 400,000 tons of ethanol. Excess ethanol can either be used for gasohol or to produce biodiesel. The production capacity of these units is approximately 400,000 tons. The country needs to export up to 80,200 tons, after which about 318,000 tons of ethanol would remain and could be used for biodiesel synthesis. Therefore, this stock is necessary to increase biodiesel production in Pakistan. In 2021, biodiesel production in the country was 0.09 thousand barrels per day; still, the country has a high feasibility of producing biodiesel in large amounts [4].

According to the Alternative Renewable Energy Policy of 2019, by 2025, Pakistan will generate 20% of its energy from renewables, and by 2030, it will generate 30%, promoting the use of alternative energy resources. In recent years, thermal energy has comprised 63%

of the energy mix, while renewable energy has suitably contributed 1.1% [60]. Contrary to the above policy, the current scenario does not contribute to either meeting the target by 2025 or reducing emissions. Furthermore, Pakistan can reach this goal sustainably by exploiting the potential of renewable energy. As a result, the biodiesel production model will support this policy, as well as waste management firms and municipalities, while taking financial limits into account. The government should also provide incentives in the form of subsidies to encourage stakeholders to participate in the execution of the program.

### 3.4. Life Cycle Cost and Economic Assessment

The economic assessment results show the viability of biodiesel production from mixed vegetable oil waste. The benefits of biodiesel production include biodiesel, material (glycerol), and metal recovery, as well as the conservation of land in terms of landfilling. Table 8 shows the revenue generation from a biodiesel production plant. Biodiesel is traded at 0.83 USD/kg, generating 2460.67 USD/day in revenue. Recovered materials and metals are traded at 0.755 USD/bag and 0.672 USD/kg, generating 224.380 USD/day and 199.721 USD/day in revenue, respectively. The total income generation by the biodiesel production plant is 1821.46 USD/day for 1 ton of processed mixed oil waste. Moreover, the biodiesel production plant produces 22 kg each month. Per day, the production cost is 2135.460 USD, and the monthly income generated by the plant is 57,796.617 USD. The yearly income generation by the plant is 638,839.631 USD. The income generated by the biodiesel plant is shown in Table 8.

**Table 8.** The production, working days, total waste processed, and income generated by the biodiesel plant.

| Product Type | Mixed Oil Waste | Material Recovery | Metal | Total |
|---|---|---|---|---|
| Total waste (kg) | 1000 | - | - | 1000 |
| Working Days | 22 | 22 | 22 | |
| Percentage in waste | 66.25 | 33.67 | 0.08 | 100 |
| Per-day production (kg) | 2534 | 2800 | 75 | 5229 |
| Per-day cost ($) | 2001.681 | 98.44258 | 35.33681 | 2135.460 |
| Per-month income ($) | 55,045.43 | 1995.776 | 755.41101 | 57,796.617 |
| Per-year income ($) | 596,443.87 | 33,428.841 | 8966.920 | 638,839.631 |

Biodiesel = 0.83 USD/kg, Material Recovery = 0.755 USD/bag, Metals= 0.672 USD/kg.

The 20 USD/ton operational cost is considered excellent, and the 3–4 year payback period is economically feasible. The operational cost of the current study is 20 USD/ton, significantly closer than that, and the PP of the initial capital investment is 4 years. As mentioned in Section 2.4, the LCC includes external and internal costs. Equation (7) was used to calculate the internal cost of biodiesel, which was calculated to be 24.33 USD/ton. The external cost was calculated using Equation (8), estimated at 3558.16 USD/ton. Therefore, the LCC of biodiesel calculated via Equation (6) was 3634.9 USD/ton. Table 8 shows the economic assessment results. Moreover, the overall economic assessment results are shown in Table 9.

**Table 9.** Results of economic assessment.

| Costs | USD/Year |
|---|---|
| Capital Costs | |
| Capital cost | 878,665.35 |
| Installation cost | 25,065.85 |
| Operation and Maintenance | |
| Maintenance cost | 3387.61 |
| Utility cost | 30,037.31 |
| Labor cost | 50,827.92 |
| Electricity cost | 19,981.72 |
| Total cost | 104,234.56 |
| Benefits | |
| Biodiesel | 596,443.87 |
| Recovery | 33,428.84 |
| Metals | 8966.920 |
| Total benefits | 638,839.631 |
| LCC (USD/ton) | 3634.9 |
| NPV | 4,648,132.82 |
| PB | 4 Years |

### 3.5. Energy Resource for Achieving Sustainable Production

A comprehensive analysis of the environmental impacts of gasoline, diesel, and biodiesel using the LCA reveals that biodiesel significantly reduces greenhouse gas emissions compared to traditional fuels. However, it also increases particulate matter (PM10) emissions, nitrous oxide, nitrogen oxides (NOx), and nutrients that contribute to eutrophication [61]. This balanced view is essential for planning a sustainable transportation system, considering both the environmental benefits and the challenges of biodiesel. Transportation companies in Malaysia need help for adopting biodiesel [62]. A differentiation strategy could help policymakers promote biodiesel usage more effectively by addressing identified barriers [63].

Consumer attitudes towards cellulosic ethanol, another renewable energy source, were explored in the United States. The survey data analysis revealed strong public support and willingness to pay more for cellulosic ethanol. This highlights the significance of consumer perceptions in the adoption of sustainable fuels [64]. The findings indicate a significant interest in alternative fuels, with the environment, energy consumption, climate change concerns, and gasoline prices being key determinants of one's willingness to pay [65,66]. These regional insights reveal the complex interplay between environmental impacts, policy challenges, and consumer attitudes in the context of sustainable biodiesel and renewable resource mobility initiatives [67].

The current study's findings provide key information about the environmental and economic aspects of biodiesel production from mixed vegetable oil waste. The conversion of mixed and different vegetable (edible and non-edible) oils in the production of biodiesel leads to huge benefits in terms of energy generated, reductions in emissions, and reductions in the amount of waste sent to landfills [68]. Biodiesel can be produced locally from a variety of feedstock, reducing our dependence on imported fossil fuels. This can enhance energy security and promote local economic development [69]. However, it also helps in achieving a circular economy and our sustainability goals.

### 4. Conclusions and Future Directions

Pakistan's energy needs can be met, and indigenous renewable energy sources in Pakistan are highly significant. Furthermore, additional research and development on renewable energies are needed to improve the effects of consumption. Considering a

country's economic and environmental conditions, this study was designed to investigate the feasibility of biodiesel production from mixed vegetable oil waste. A medium-scale 1 t (1000 kg) biodiesel plant was designed, and from 1000 kg of mixed vegetable oil waste, 400 kg of biodiesel can be produced. Pakistan can address its energy supply disparities by effectively implementing biodiesel in energy production. This would require supplying energy for household cooking, powering vehicles, and supporting industrial processes, including electricity generation. Nonetheless, a more comprehensive and thoughtful approach to research is needed to promote renewable energy technologies and establish clear biodiesel policies for the government. This should not be marginal but rather a deliberate focus on strengthening initial local research initiatives. Our research highlights the significance of aligning with the United Nations' SDGs. In particular, our work contributes to the progress of SDG-7, which stresses the importance of accessible and clean energy, and SDG-12, which promotes responsible consumption and production.

An LCA was performed to estimate the current project's environmental impacts. The functional unit was 1 t. Thus, three steps (classification, characterization, and normalization) were performed. In addition, midpoint and endpoint assessments were also conducted. The calculated midpoint impacts were CC: $1.36 \times 10^{-5}$ kg $CO_2$ eq, HT: 5.31 kg 1.4 DB eq, OD: 0.00271 kg CFC-11 eq, AP: 123 kg $SO_2$ eq, and POF: 51.4 kg $NO_x$ eq. To determine the main contributors' stages in the life cycle of biodiesel, the relative contribution by individual biodiesel type was calculated. The percentage share of ecotoxicity is greater and has an impact, particularly on marine ecotoxicity and human toxicity. Thus, its collection and transportation at plants show that usage and transportation are leading stages. This process is more fuel-intensive than other processes. To further alleviate the impacts, scenario modelling was conducted, in which the electricity supply was from photovoltaic solar cells. As a result, the global warming potential increases from $1.36 \times 10^{-5}$ to $2.91 \times 10^{-5}$ kg $CO_2$ eq., and the fine particle matter formation and freshwater ecotoxicity also decrease from 44.5 to 0.725 kg $PM_{2.5}$ eq. and 0.647 to 0.136 kg 1.4 DB eq., respectively. Furthermore, the effect on freshwater consumption in the current model is 326 $m^3$, which decreased to 279 $m^3$ in the scenario model. Likewise, human toxicity (cancer) and marine eutrophication decreased from 5.31 to 0.405 kg 1.4 DB eq. and 0.134 to 0.00765 kg N eq., respectively. The economic analysis showed that biodiesel is traded at 0.83 USD/kg, generating 2460.67 USD/day (753,132.84 PKR/day). Recovered materials and metals are traded at 0.755 USD/bag and 0.672 USD/kg, generating 224.380 USD/day and 199.721 USD/day in revenue, respectively. Furthermore, the total income generated by the biodiesel plant is 1821.46 USD per day (0.6 million PKR/day) for 100 tons of processed mixed vegetable oil waste. The yearly income generated by the plant is 0.6 million USD (195 million PKR). The payback period of the initial capital investment is four years.

**Author Contributions:** Conceptualization, F.M.; methodology, K.S.; data curation, I.A.; review and editing, A.-S.N.; software, M.H.J. and A.A.; final editing, M.N. All authors have read and agreed to the published version of the manuscript.

**Funding:** This research received no external funding.

**Institutional Review Board Statement:** Not applicable.

**Informed Consent Statement:** Not applicable.

**Data Availability Statement:** No new data were created or analyzed in this study.

**Conflicts of Interest:** The authors declare no conflict of interest.

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
