# Peer review of "Advancing Biodiesel Production System from Mixed Vegetable Oil Waste: A Life Cycle Assessment of Environmental and Economic Outcomes"

_sustainability, doi:10.3390/su152416550_

Round 1

Reviewer 1 Report

Comments and Suggestions for Authors

The study provides a life cycle assessment of bio-oil production alongside economic and environmental evaluations. The authors have conducted a commendable job, and a few suggestions for enhancing the study are proposed:

Include a reference to the source at Line 48-49.

Line 181 - When FFAs are mentioned for the first time, it is better to decipher the abbreviation

Line 396-400 Is the calorific value given per 1 kg or 1 g? Values around 37 kJ/kg are very low.

Figures 5a-d Include error bars where possible.

Author Response

First, we sincerely thank all reviewers and the editor for the time spent reading and commenting on this manuscript and your insightful and valuable comments. We found your feedback very constructive and valuable. We believe the revised manuscript is much stronger and improved with your corrections. We sincerely hope that we have properly addressed your comments in the answers below and with our edits to the revised manuscript. We hope that in its present form, the presentation of the paper is now satisfactory and up to a high publishable standard. All the changes and some improvements made in the revised manuscript. We have edited, added, and expanded many sections based on the reviewers' comments and made significant corrections in the revised manuscript.

Comments from Editors and Reviewers:

  1. The final part of the abstract, you can clearly indicate that you support the development of SDGs 7 and 12 (free to choose but I looking at the paper they seem to stand out to me) or others that you want to highlight.

Answer: Thank you so much for your valuable and constructive suggestion. Accordingly, final part of the abstract has been changed.

  1. Section 1 is missing two three sentences on the SDGs with references. Your source 23 indicates several SDGs that support this resource. However, my suggestion is to narrow it down. There is also a need to improve the style in which they are presented e.g. SDG 7 (Affordable and Clean Energy). In addition, I think the section editorial is useful to consider in explaining the novelty of the work https://www.mdpi.com/2071-1050/15/12/9443.

Answer: Thank you for the constructive comment. We have added two three sentences related to specific SDGs (SDG-7 and SDG-12).

  1. Before section 2 and 3 propose a brief description of how the sections will be divided into the various subsections this helps the reader to keep the thread.

Answer: Thank you so much for your valuable comment and constructive suggestion. For the convenience of readers, we have added brief descriptions before sections 2 and 3.

  1. At the end of section 3 I suggest a brief comparison with existing literature highlighting for example the role: "energy resource for achieving sustainable production" and "the Diffusion of Sustainable Mobility Initiatives: A Stakeholder Attitudes Assessment". Biodiesel can make a great contribution in combination with other renewable resources.

Answer: Thank you so much for the constructive suggestion. Accordingly, we have added a sub-section at the end of section 3 that defines biodiesel as great contribution to renewable energy transition.

Reviewer 2 Report

Comments and Suggestions for Authors

The work compares its results with those of other authors. For instance, in Table 3, a summary of different studies conducted on the environmental impact assessment of the life cycle of biodiesel production is presented. Additionally, various studies and references are cited throughout the document.

The analysis of generating energy from discarded vegetable oils through biodiesel production presents a sustainable and economically viable approach. This process benefits the environment and contributes to cost savings by reducing waste disposal.

Regarding the presentation of information, there are some details, such as:

-The title of line 233 lacking numbering.

-In tables, as well as the rest of the document where figures are presented, the criteria for decimal usage are not standardized.

-Similarly, the use of the comma as a thousands separator is not standardized.

- The document covers the topic of bioethanol in line 599.

Regarding the substance of the information, the following suggestions are made:

-the presented results are very good; however, it does not explain if regulations allow the development of such projects and the commercialization of the generated biodiesel.

-There is no discussion of the production cost results of biodiesel compared to other research findings, both in the same country and globally, to provide a broader perspective on the project's viability.

-The conclusions are heavily laden with results, so they should be more concise.

Author Response

(The authors gave the same response as above.)
